# Cell-free DNA in spent culture medium effectively reflects the chromosomal status of embryos following culturing beyond implantation compared to trophectoderm biopsy

**Akihiro Shitara** *, **Kazumasa Takahashi, Mayumi Goto, Harunori Takahashi, Takuya Iwasawa, Yohei Onodera, Kenichi Makino, Hiroshi Miura, Hiromitsu Shirasawa, Wataru Sato, Yukiyo Kumazawa, Yukihiro Terada**

Department of Obstetrics and Gynecology, Akita University Graduate School of Medicine Hondo, Akita, Japan

* shitaraa@med.akita-u.ac.jp

## Abstract

This prospective study evaluated the accuracy of non-invasive preimplantation genetic testing for aneuploidy (niPGT-A) using cell-free DNA in spent culture medium, as well as that of preimplantation genetic testing for aneuploidy (PGT-A) using trophectoderm (TE) biopsy after culturing beyond implantation. Twenty frozen blastocysts donated by 12 patients who underwent IVF at our institution were investigated. Of these, 10 were frozen on day 5 and 10 on day 6. Spent culture medium and TE cells were collected from each blastocyst after thawing, and the embryos were cultured *in vitro* for up to 10 days. The outgrowths after culturing beyond implantation were sampled and subjected to chromosome analysis using next-generation sequencing. Chromosomal concordance rate, sensitivity, specificity, positive predictive value (PPV), negative predictive value (NPV), false-positive rate (FPR), and false-negative rate (FNR) of niPGT-A and PGT-A against each outgrowth were analyzed. The concordance rate between the niPGT-A and outgrowth samples was 9/16 (56.3%), and the concordance rate between the PGT-A and outgrowth samples was 7/16 (43.8%). NiPGT-A exhibited 100% sensitivity, 87.5% specificity, 88.9% PPV, 100% NPV, 12.5% FPR, and 0% FNR. PGT-A exhibited 87.5% sensitivity, 77.8% specificity, 87.5% PPV, 75% NPV, 14.3% FPR, and 22.2% FNR. NiPGT-A may be more accurate than PGT-A in terms of ploidy diagnostic accuracy in outgrowths.

## Introduction

Human embryos are prone to chromosomal abnormalities, primarily due to chromosome separation errors occurring during meiosis [1]. Chromosomal abnormalities can cause miscarriages in early pregnancy and many serious chromosomal disorders [2]. The occurrence of

**Data Availability Statement:** All relevant data are within the manuscript and its Supporting Information files.

**Funding:** · Y.T · Grant Number 18H02942 · JSPS KAKENHI · https://www.jsps.go.jp/english/index.html · The funders had no role in study design, data collection and analysis, decision to publish, or preparation of the manuscript.

**Competing interests:** The authors have declared that no competing interests exist.

chromosomal abnormalities is significantly higher in older mothers, patients who experience recurrent miscarriages, or those with chromosomal abnormalities such as translocations that result in poor clinical outcomes in reproductive medicine.

Studies have reported that assisted reproductive technology (ART) outcomes can be improved by embryo selection based on preimplantation genetic testing for aneuploidy (PGT-A) [3]. However, PGT-A offers no significant improvement in fertility rates or reduction in miscarriage rates [4]; additionally, no consensus on its effectiveness has been reached to date. Moreover, recent studies have reported that PGT-A performed in combination with next-generation sequencing (NGS) leads to an increase in the detection of chromosome mosaicism in trophectoderm (TE) biopsies. Mosaic embryos may result in healthy births [5, 6]; however, they are associated with lower implantation rates compared to euploid embryos [7, 8]. In a previous study, we found a discrepancy between the chromosomal states of cells in TE biopsies and the whole embryo, and the results of TE biopsies do not necessarily represent the whole embryo (unpublished data).

Additionally, there is concern that the TE biopsy may be invasive for the embryo. In the general PGT-A method, approximately five to ten cells from the TE in the blastocyst stage are biopsied and analyzed [9, 10], which is susceptible to the quality of the biopsy technique. Although biopsy-induced embryo loss is estimated to be less than 10%, some laboratories have reported biopsy embryo loss as high as 30% [11]. The future developmental risk to embryos that underwent TE biopsies is also controversial [12, 13]. Some researchers have proposed a non-invasive preimplantation genetic testing for aneuploidy (niPGT-A) that analyzes a spent culture medium. In particular, Xu *et al.* demonstrated a high chromosomal concordance rate between spent culture medium and their corresponding embryos [14]. If sufficiently accurate and reproducible, these methods could be attractive alternatives to TE biopsies for PGT-A. One study found that niPGT-A produced by a spent culture medium was less likely to cause errors related to embryo mosaicism and was more reliable than TE biopsy PGT-A [15]. Due to a high DNA amplification failure rate and a low chromosomal concordance rate, spent culture medium are not considered optimal for PGT-A, and the niPGT-A operation has not been adequately determined to date [16].

The newly established embryo culturing system has enabled embryo culturing beyond implantation up to 14 days *in vitro* [17, 18]. This embryo culturing system offers the opportunity to investigate chromosomal instability after early post-implantation development. Using this embryo culturing protocol, we conducted an experiment to examine how TE biopsy and spent culture medium reflect the embryo chromosomal status in the early post-implantation period up to day 10. Although there is still debate regarding the consensus on the clinical management of PGT-A and niPGT-A, assessing chromosomes during embryogenesis will provide new insights into embryo selection in clinical practice. Furthermore, we examined the relationship between chromosomal status in TE biopsy and chromosomal status in spent culture medium and embryonic growth. These findings can be expected to provide useful information for reproductive medicine research.

## Material and methods

### Ethical approval

This study was approved by the Ethics Committee of Akita University (Permission number: 1090.2) and the Japan Society of Obstetrics and Gynecology (Permission number: 127). The embryos used in this study were used after obtaining informed consent from each patient. The donated embryos were handled according to the Japan Society of Obstetrics and Gynecology policy regarding research using human sperm, ova, and fertilized eggs.

## Embryo source and background

This study used 20 surplus embryos recovered from 12 infertile patients at Akita University Hospital from 2006–2016. Embryos were used after anonymization and acquisition of the written informed consent of each donor. We used embryos that were not used clinically and marked for disposal. Of the targeted embryos, ten were frozen on day 5 and ten on day 6. The mean patient age (± standard deviation, SD) at the time of embryo freezing was 35.6 ± 3.2 years, and the average number of egg collections was 1.6 ± 0.9. The qualitative grade of embryos, when frozen, was evaluated by applying the Gardner system of classification [19].

## Embryo thawing and TE biopsy/spent culture medium collection

Thawing of embryos was performed using the Cryotop Safety Kit (Kitazato, Japan) according to the manufacturer's instructions. The thawed embryos were subjected to recovery culturing at 37˚C, using 6% $CO_2$ and 5% 02 Sequential Blast™ medium (ORIGIO, Denmark). Embryos were briefly exposed to acidic Tyrode (Kitazato, Japan) to remove the zona pellucida prior to TE biopsy. After removing the zona pellucida, 5-day-old embryos were cultured in blast medium for 24 hours, and 6-day-old embryos were cultured in blast medium for 3 hours.

Biopsy of TE cells was performed on day 6 in all embryos. Five to ten cells were collected using the flick method through a biopsy pipette (ORIGIO, Denmark) in a modified HTF culture medium with HEPES, HSA (Kitazato, Japan). The collected TE cell mass was immersed in new mHTF, 1% PVP (7% polyvinylpyrrolidone (PVP) solution with HSA (Irvine Scientific, USA) in DPBS (-) (Thermo Fisher Scientific, USA) diluted to 1%), and stored frozen in 2.5 μL DPBS in a polymerase chain reaction (PCR)-use tube at −20˚C until analysis. During TE cell biopsy, the spent culture medium used for culturing after removing zona pellucida was collected for niPGT-A and cryopreserved. In addition, all equipment used was replaced for each sample.

## Embryo culturing beyond implantation

After performing the TE biopsy, embryo culturing beyond implantation was performed according to the protocol presented in previous studies [17, 18, 20]. The embryos were transferred to a μ-Dish 35mm, low (ibidi, Germany) filled with IVC1 medium (Cell Guidance Systems, Cambridge). Half of the culture medium was replaced every 24 hours. The embryos were attached on day 7 or 8, following which the medium was replaced with IVC2, and half of the culture medium was replaced every 24 hours. Further, based on a previous study, 100 ng/mL of recombinant human/murine/rat activin A (Peprotech, USA) was added to the culture medium during the culturing beyond implantation (days 6–10); cultures were performed under hypoxic conditions [20–22].

## Outgrowth sampling

Long-term culturing was performed up to day 10. The embryos were evaluated morphologically every 24 hours. Embryos that did not develop compared to the previous day or embryos that were once attached but became detached were sampled. Embryos that stopped growing on day 8 or 9 were difficult to biopsy; consequently, the whole embryos were sampled. Embryos that exhibited sufficient growth up to day 10 were detached from the culture dish using a Bio-Cut Blade℞ 15˚ (Feather, Japan), and the center of the embryo was biopsied and sampled. The collected sample was frozen and stored in a PCR tube at −20˚C until analysis, as in the TE biopsy method. Further, all equipment used was replaced for each sample.

### Next-generation sequencing

Sample cell preparation, cell lysing procedures, DNA extraction, and whole genome amplification (WGA) were performed using the SurePlex DNA Amplification System (Illumina, San Diego, CA, USA), according to the manufacturer's recommended conditions. The DNA concentration of the product after amplification was measured using a Qubit 3.0 Fluorometer (ThermoFisher Scientific) with the Qubit dsDNA HS Assay kit (ThermoFisher Scientific).

Following WGA, the samples were treated using a VeriSeq PGS Kit (Illumina, San Diego, CA, USA), according to the manufacturer's instructions. NGS was performed using a MiSeq testing device (Illumina, San Diego, CA, USA). The obtained data were analyzed using Blue-fuse Multi software to obtain the karyotype information of the sample.

### Pilot study

Culturing beyond implantation was conducted in advance on a total of five embryos according to the culture protocol described above. In a previous report, the developmental progress rate for the period up to day 12 was found to be 38/73 (52%) (20), but at our facility, no favorable growth was observed until day 12. Four of five embryos developed favorably up to day 10. During the Carnegie stage, epiblasts and amniotic cavities were already formed by day 10 [23], and we decided to limit the culture period to ten days in this experiment.

### Statistical analysis

Statistical analysis was performed using the R statistical software v3.6.1 (R Foundation for Statistical Computing). Statistical significance was evaluated by performing McNermar's $x^2$ test or finding the Kappa coefficient. $P$-values less than 0.05 were considered to be statistically significant.

## Results

This study evaluated embryos with unknown chromosomal profiles. From these blastocysts, TE biopsy cells, i.e., PGT-A samples (n = 20); spent culture medium on day 6, i.e., niPGT-A samples (n = 20); and embryo after culturing beyond implantation, i.e., outgrowth samples (n = 20) were obtained. The study flow chart is shown below. (Fig 1) NGS analysis was performed after each WGA procedure, and 55 of 60 samples were analyzed. Results were obtained for 19/20 PGT-A (95%), 19/20 niPGT-A (95%), and 17/20 outgrowth samples (85%). Samples that could not be analyzed were considered to be unanalyzable due to excessive noise. The average DNA concentration after WGA was 41.4 ng/μL (34.0–49.8 ng/μL), 22.2 ng/μL (9.3–32.8 ng/μL), and 34.8 ng/μL (4.3–44.4 ng/μL) for PGT-A, niPGT-A, and outgrowth samples, respectively. Although the culture time after removing the zona pellucida was different between the 5-day-old and 6-day-old embryos, there was no difference in the DNA concentration of the culture medium collected for niPGT-A (5-day-old embryo group: 22.0ng/μL vs. 6-day-old embryo group 19:22.3 ng/μL; $p = 0.807$).

Based on the results, each sample was classified into one of the following three categories: euploid, aneuploid (copy number exhibits 80% or more aneuploidy), or mosaic (copy number exhibits 20–80% aneuploidy). Table 1 shows the breakdown of each analysis result. A total of 60 analysis results were obtained from the PGT-A, niPGT-A, and outgrowth samples, and they were subsequently compared. (Table 2)

Favorable development was observed in 10/20 (50%), 5/20 (25%), and 5/20 (25%) embryos up to days 10, 9, and 8, respectively. When the embryos were divided into euploid and abnormal (aneuploid or mosaic) groups, among those that developed until day 10, 7/9 (77.8%), 8/10

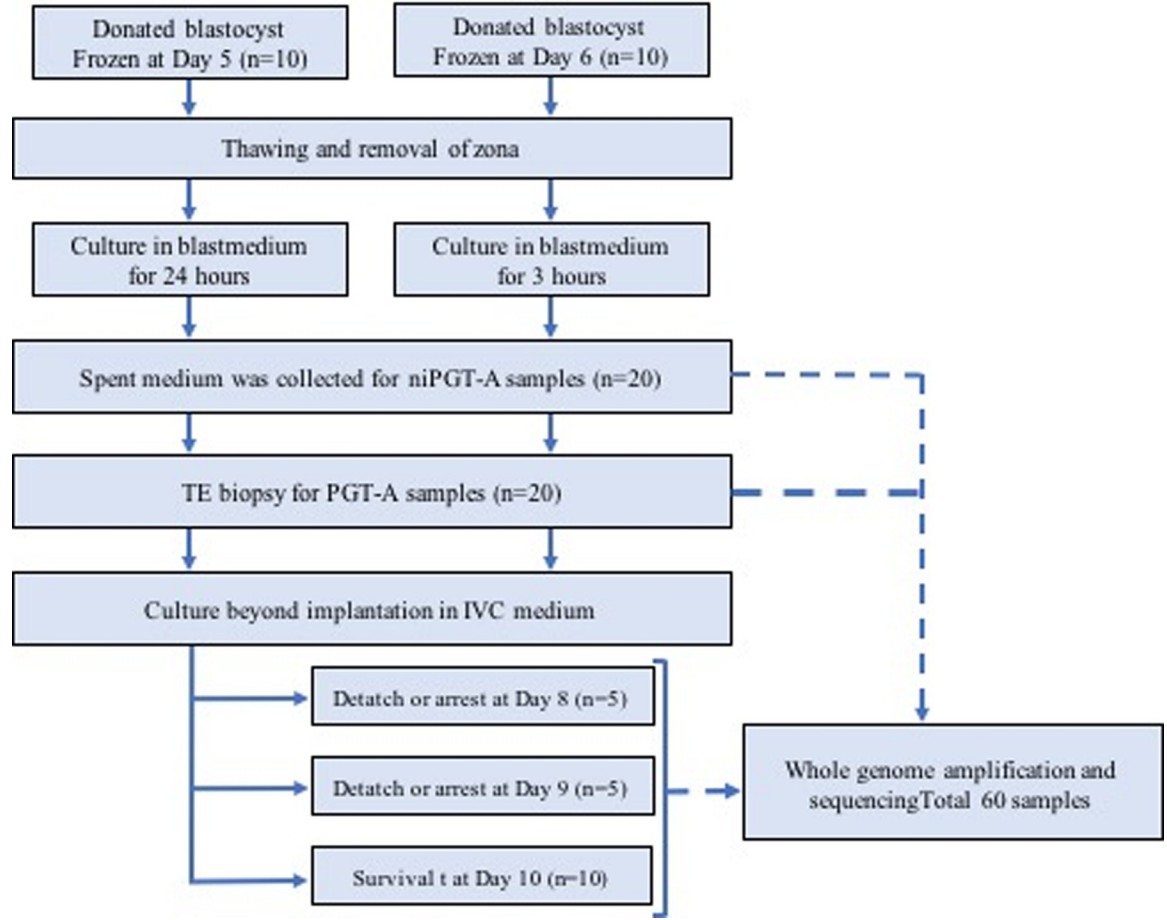

**Fig 1. Study flow chart.** A total of 20 blastocysts were analyzed. TE biopsy was performed for PGT-A on all embryos on day 6. Spent culture medium used for culture after removing zona pellucida was collected for niPGT-A. Culturing beyond implantation was subsequently performed, and five 8-day-old embryos, five 9-day-old embryos, and ten 10-day-old embryos were sampled. A total of 60 samples were analyzed using next-generation sequencing.

(80%), and 7/10 (87.5%) of the PGT-A, niPGT-A, and outgrowth samples, respectively, exhibited a chromosome analysis result of euploidy. Moreover, among the embryos that developed until day 10, there were chromosomal abnormalities in 2/9 (22%), 2/10 (20%), and 3/10 (30%)

**Table 1. Breakdown of euploid/aneuploidy/mosaic embryos and the feasibility rate of each determinant.**

|  | PGT-A | niPGT-A | Outgrowth |
|---|---|---|---|
| Number of analyzed embryos | 19 (95%) | 19 (95%) | 17 (85%) |
| Failed amplification | 1 (5%) | 1 (5%) | 3 (15%) |
| Euploid | 9 (47%) | 10 (53%) | 8 (47%) |
| Aneuploid[*1] | 5 (26%) | 4 (21%) | 2 (11%) |
| Single | 3 (16%) | 3 (16%) | 0 |
| multiple | 2 (11%) | 1 (5%) | 1 (6%) |
| Mosaic[*2] | 5 (26%) | 5 (26%) | 7 (41%) |

Twenty embryos were used; 20 PGT-A samples, 20 niPGT-A samples, 20 outgrowth samples. A total of 60 samples were analyzed.

[*1] copy number exhibits aneuploidy of 80% or more [*2] copy number exhibits 20–80% aneuploid

**Table 2. Comparison of results from 60 samples obtained from 20 embryos.**

| No | Time of freezing (day) | Age | Grade | Time of Outgrowth collection (day) | PGT-A | Outgrowth | niPGT-A |
|---|---|---|---|---|---|---|---|
| 1 | 5 | 35 | 4BB | 10 | 47XX,+14 | 46XX | 46XX |
| 2 | 5 | 29 | 3BA | 9 | 46XX,-8(80%) | 46XX,8p(del) | 46XX,-8(80%), +22(80%) |
| 3 | 5 | 36 | 4AA | 10 | 46XX | 46XX | 46XX |
| 4 | 5 | 36 | 3AA | 10 | 46XY | 46XY,+19(60%) | 46XY |
| 5 | 5 | 40 | 4BB | 9 | 46XY,+15(80%), +22(80%) | N/A | 47XY,+15,+22(80%) |
| 6 | 5 | 40 | 3BB | 8 | 46XX,+16,-10 | 46XX,+16,-10 | 46XX,+16,-10 |
| 7 | 5 | 36 | 3AA | 10 | 46XX | 46XX | 46XX |
| 8 | 5 | 36 | 4AB | 10 | 45XY,-14, +13(80%),+15 (80%) | 46XY,+3,-14 | 45XY,-14 |
| 9 | 5 | 36 | 4AB | 10 | 46XX | 46XX,3q(dup),-20(60%) | 46XX,+1(60%),+4(60%),-5 (60%),+15(80%), |
| 10 | 5 | 40 | 4BB | 8 | 45XO,-14(60%) | 45XX,-14,+2(80%),-10(80%),+18 (80%), +20(80%) | 46XO,+14 |
| 11 | 6 | 33 | 4AA | 8 | 46XY | 46XY | 46XY |
| 12 | 6 | 35 | 5BA | 10 | 46XY | 46XY | 46XY |
| 13 | 6 | 34 | 4BA | 9 | 47XY,+15,+6(60%),+12 (60%),+16(80%) | 46XY,-9(60%) | N/A |
| 14 | 6 | 34 | 5AB | 8 | 46XY | N/A | 46XY |
| 15 | 6 | 33 | 4BB | 9 | 45XX,-15 | 46XY,-9(60%) | 45XX,-15,-9(80%) |
| 16 | 6 | 37 | 5AA | 10 | N/A | 46XX | 46XX |
| 17 | 6 | 37 | 3BB | 9 | 45XX,-11 | 45XX,-11,+14(60%) | 45XX,-11 |
| 18 | 6 | 38 | 3AA | 10 | 46XX | 46XX | 46XX |
| 19 | 6 | 38 | 3AA | 8 | 48XX,+8,+21 | N/A | 47XY,+8,-1(60%), |
| 20 | 6 | 38 | 5BB | 10 | 46XY | 46XY | 46XY |

of PGT-A, niPGT-A, and outgrowth samples, respectively. These results also suggest that embryos with chromosomal abnormalities exhibit poor development.

Next, we examined the autosomal concordance rate for the outgrowth of PGT-A and niPGT-A samples for each embryo (Table 3). Only samples that were identical with respect to the extent of polyploidy and mosaicism were considered as concordant, and partial concordance was considered as a discordance. The concordance rate between the PGT-A and outgrowth samples was 7/16 (43.8%), and that between the niPGT-A and outgrowth samples was 9/16 (56.3%). There was no significant difference in the concordance rate between the PGT-A and niPGT-A samples (p = 0.7244). Notably, there was one case where the PGT-A and outgrowth samples did not match, while the niPGT-A and outgrowth samples matched.

**Table 3. Chromosomal concordance rate.**

| | Autosomal chromosome concordance (%) | Sex chromosome concordance (%) |
|---|---|---|
| PGT-A vs. Outgrowth | 7/16(43.8) | 14/16(87.5) |
| niPGT-A vs. Outgrowth | 9/16(56.3) | 14/16(87.5) |
| PGT-A vs. niPGT-A | 10/18(55.6) | 17/18(94.4) |
| PGT-A vs. niPGT-A vs. Outgrowth | 7/15(46.7) | 13/15(86.7) |

This table shows the autosomal concordance rate and the sex chromosome concordance rate.

There were no cases where the niPGT-A and outgrowth samples did not match, while the PGT-A and outgrowth samples matched. The concordance rate between the PGT-A and niPGT-A samples was 10/18 (55.6%).

The concordance rate between sex chromosomes in the PGT-A and outgrowth samples was 14/16 (87.5%), whereas that between the niPGT-A and outgrowth samples was 14/16 (87.5%). The PGT-A and niPGT-A samples were 17/18(94.4%) in concordance.

In terms of the relationship between embryo development and concordance rate, in 10-day-old embryos, the concordance rate between the PGT-A and outgrowth samples was 5/9 (55.6%), whereas that between the niPGT-A and outgrowth samples was 7/10 (70%). In embryos that stopped developing on day 8 or 9, the concordance rate between PGT-A and outgrowth samples was 2/7 (28.6%), and that between the niPGT-A and outgrowth samples was 2/6 (33.3%) (Table 4). Based on these results, although the autosomal concordance rate was not significant, this rate tended to be higher in niPGT-A samples.

We calculated sensitivity (i.e., true positive), specificity (i.e., true negative), positive predictive value (PPV), negative predictive value (NPV), false-positive rate (FPR), and false-negative rate (FNR) to estimate the diagnostic accuracy of PGT-A and niPGT-A samples with respect to outgrowth chromosomes (Table 5).

For sensitivity (i.e., when outgrowth was abnormal), the probability that a PGT-A sample (or niPGT-A sample) would also be abnormal was 87.5% and 100% in the PGT-A and niPGT-A groups, respectively.

For specificity (i.e., when outgrowth was euploid), the probability that a PGT-A sample (or niPGT-A sample) would also be euploid was 77.8% and 87.5% in the PGT-A and niPGT-A groups, respectively.

For PPV (i.e., when PGT-A samples (or niPGT-A samples) exhibited abnormal results), the probability of abnormal outgrowth was 87.5% and 88.9% in the PGT-A and niPGT-A groups, respectively.

For NPV (i.e., when PGT-A samples (or niPGT-A samples) were euploid), the probability that outgrowth was also euploid was 75% and 100% in the PGT-A and niPGT-A groups, respectively.

For FPR (i.e., when PGT-A samples (or niPGT-A samples) were abnormal), the probability that outgrowth was also euploid was 14.3% and 12.5% in the PGT-A and niPGT-A groups, respectively.

For FNR (i.e., when PGT-A samples (or niPGT-A samples) were euploid), the probability of abnormal outgrowth was 22.2% and 0% in the PGT-A and niPGT-A groups.

For diagnostic accuracy, the kappa statistic of the PGT-A and niPGT-A groups was 0.62 (95% CI: 0.25–1) and 0.77 (95% CI: 0.47–1), respectively. Although not significantly higher, the niPGT-A group exhibited a high degree of concordance.

Based on these results, the niPGT-A group may have a higher diagnostic accuracy than the PGT-A group.

**Table 4. Comparison of autosomal concordance rate between day 10 and day 8 or day 9.**

|  | day 10 (%) | day 8 or 9 (%) |
|---|---|---|
| PGT-A vs. Outgrowth | 5/9(55.6) | 2/7(28.6) |
| niPGT-A vs. Outgrowth | 7/10(70) | 2/6(33.3) |
| PGT-A vs. niPGT-A | 6/9(66.7) | 4/9(44.4) |
| PGT-A vs. niPGT-A vs. Outgrowth | 5/9(55.6) | 2/6(33.3) |

This table shows the autosomal concordance rate between 10-day-old and 8-day-old or 9-day-old embryos.

**Table 5. Diagnostic accuracy of PGT-A and niPGT-A groups.**

|  | PGT-A | niPGT-A |
|---|---|---|
| sensitivity | 87.5% | 100% |
| specificity | 77.8% | 87.5% |
| PPV | 87.5% | 88.9% |
| NPV | 75% | 100% |
| FPR | 14.3% | 12.5% |
| FNR | 22.2% | 0% |

PPV: positive predictive value; NPV: negative predictive value; FPR: false-positive rate, FNR: false-negative rate

These calculations were done with respect to outgrowth chromosomes

## Discussion

In this study, culturing beyond implantation was used to obtain new cytogenetic findings of PGT-A/niPGT-A. All our analyses were TE biopsied at day 6 and cultured up to day 10. Spent culture medium of the recovery culture from embryo thawing to TE biopsy was collected for niPGT-A, and the results of TE biopsies of PGT-A samples were correlated with the outgrowth chromosomal status, respectively. To the best of our knowledge, this is the first study to investigate the relationship between embryos after culturing beyond implantation and PGT-A or niPGT-A.

The report of chromosomal mosaicism using NGS has been a subject of intense debate regarding the diagnostic accuracy of PGT-A, especially the predictive value of TE biopsy to reflect the chromosomal status of the whole embryo. Various studies have reported PGT-A results obtained using TE biopsy, but the concordance rate between TE and ICM karyotypes is approximately 62.1–86.2% [24–26]. Additionally, Ou et al. re-examined embryos considered abnormal using PGT-SR and reported a concordance rate of 68% between TE biopsies and total blastocysts [27]. Another study conducted at our institution also found that the TE biopsy and total blastocysts had a chromosomal concordance rate of 67.7%. Therefore, the results of TE biopsies do not accurately represent the chromosomal status of the entire human blastocyst. In the present study, the concordance rate between the PGT-A and outgrowth samples was 43.8%, which is low compared to previous reports of PGT-A. Embryos after culturing beyond implantation are considered closer to the post-implantation chromosomal status than the blastocyst stage; thus, this result also suggests that the diagnostic rate of PGT-A is not accurate. To date, the only study on embryos after culturing beyond implantation and PGT-A was the one conducted by Popovic *et al.* [20], who reported the diagnostic accuracy of PGT-A via TE biopsy to be approximately 80%. Although our study applied the protocol described in their report, the results were not similar. This is possibly because favorable embryos of 5BB or more were analyzed by Popovic *et al.*, whereas in the current study, only four embryos of 5BB or more were available, and 16 embryos were graded 5BB or less. In the group of embryos above 5BB, 3/4 PGT-A results were euploid (one was not determined), and 3/4 outgrowth results were also euploid. Favorable development was observed even in the case of inferior grades such as embryos No. 7 and No. 18, and while there are cases where outgrowth karyotype is euploid, the concordance rate between the PGT-A and outgrowth samples was 5/14 (35.7%) in embryos graded 5BB or lower.

If PGT-A by TE biopsy does not truly reflect the chromosomal status of the blastocyst, invasive procedures on the embryo should be avoided. Embryo biopsies have been shown to reduce embryo quality at the cleavage stage [28] and may be detrimental to their development and implantation [29]. In animal studies, embryo biopsy has been found to affect neurodevelopment

and adrenal development, but the evaluation of long-term prognosis in humans remains incomplete [12, 13].

The introduction of niPGT-A in clinical practice is relatively easy because it does not require the technology related to embryo biopsy. However, one problem is the diagnostic accuracy of niPGT-A. Capalbo *et al.* reported that the concordance rate between spent culture medium samples and TE biopsies was low at 20.8% for PGT-M and did not recommend using spent culture medium for PGT-M [16]. Ho *et al.* reported that the chromosomal concordance rate between spent culture medium on day 3 and the whole embryo was 56.3%, and chromosomal concordance rate between spent culture medium on day 5 and the TE biopsy was 65% [30]. Xu *et al.* reported achieving high sensitivity between the spent culture medium and blastocysts through multiple annealing and looping-based amplification cycles (MALBAC)-WGA [14]. Jiao *et al.* reported achieving a 90% clinical concordance rate between spent culture medium and blastocysts following modifications to the MALBAC method [31]. In our study, the concordance rate between niPGT-A samples and outgrowth was 56.3%, which was higher than that of PGT-A samples.

Notably, the FNR for the niPGT-A samples was 0% in our study. In previous studies, a low FNR for niPGT-A has been reported. The FNR of niPGT-A for TE biopsy reported by Huang *et al.* was 0% [15], and the FNR for niPGT-A for blastocysts reported by Xu *et al.* was 11.8% [14]. In addition to the low FNR, our results also demonstrated that the niPGT-A samples were superior in terms of sensitivity and specificity. A low FNR reduces the possibility of transplanting an embryo that is unsuitable for transplantation and may help improve clinical outcomes.

In this study, the general NGS protocol was used. Accordingly, although the results of this study may lead to the clinical application of niPGT-A, it is necessary to minimize measurement noise and DNA amplification failure. Yeung *et al.* reported that 150/168 samples (89.3%) were successfully amplified and sequenced [32]. The determinable average DNA concentration following WGA was 20.0 ng/μL (2.7–59.6 ng/μL), and the average for our niPGT-A samples (22.2 ng/μL, 9.3–32.8 ng/μL) was almost the same. As shown in the results, there was no difference between the cfDNA concentration of 5-day-old embryos and that of 6-day-old embryos. However, this is significantly lower (p <0.01) when compared to the average DNA concentration of PGT-A samples (41.4 ng/μL). In the present study, while it was possible to determine 19/20 results for both the PGT-A and niPGT-A samples, this difference in DNA concentration may lead to a measurement error. In the study mentioned above conducted by Xu *et al.*, ploidy information in 100% of 42 samples could be obtained using MALBAC-WGA [14]. NiPGT-A has the potential to become an alternative to TE biopsy if an optimized method can be developed.

The origin of cell-free DNA is questionable, but embryonic DNA from apoptotic cells may contribute. In the embryos during culturing, both ICM and TE undergo apoptosis [33, 34]; consequently, DNA in the spent culture medium may be derived from both these cell lines. Zhu *et al.* reported that many aneuploid cells underwent apoptosis following elimination from the embryo [35]. In addition, Hashimoto *et al.* used mouse models to verify if ICM-constituting cells were selectively culled by heterogeneous cells due to cellular competition at the epiblast formation stage, which then developed into the fetus [36]. However, some euploid cells also undergo apoptosis, and euploid results cannot be obtained unless DNA is discharged into the spent culture medium. For euploid embryos, leakage of DNA from euploid cells is thought to exceed the extent of discharge from aneuploid cells; however, further studies are warranted in this area.

Bolton *et al.* conducted a study using mosaic mouse models, demonstrating that both aneuploid and euploid cells undergo apoptosis from ICM and TE, thereby indicating that a higher

proportion of ICM cells undergo apoptosis compared to TE cells [37]. This result indicates that TE biopsies reflect only the chromosomal status of TE, whereas the cell-free DNA in the spent culture medium reflects the chromosomal status from both ICM and TE. Thus, the spent culture medium reflects the overall condition of blastocysts more clearly than TE biopsies.

This study has some limitations; the embryos that could be used were limited. In addition, it was not possible to clarify whether this method expresses the original attributes of embryos after implantation.

## Conclusions

In summary, the results of this study may support the application of niPGT-A using cell-free DNA in spent culture medium. The high chromosomal concordance rate, sensitivity, specificity, as well as FNR of 0% all suggest that niPGT-A may be superior to PGT-A. If the niPGT-A methodology can be perfected, it may be possible to perform preimplantation screening more easily while avoiding the potentially adverse effects of invasive procedures on the embryo. It remains desirable to develop better WGA methods and accumulate additional research data.

## Supporting information

**S1 Fig. All NGS results of embryos (PGT-A, outgrowth, niPGT-A).** NGS was performed using a MiSeq testing device (Illumina, San Diego, CA, USA). The obtained data were analyzed using Bluefuse Multi software to obtain the karyotype information of the sample. https://doi. org/10.6084/m9.figshare.13488771.
(PDF)

**S1 Table. Results obtained from all samples.** This table shows the overview of chromosomal profiles for all embryos used in the study. https://doi.org/10.6084/m9.figshare.13488762.
(XLSX)

**S2 Table. DNA concentration after WGA for PGT-A, outgrowth, niPGT-A.** Whole genome amplification (WGA) were performed using the SurePlex DNA Amplification System (Illumina, San Diego, CA, USA), according to the manufacturer's recommended conditions. The DNA concentration of the product after amplification was measured using a Qubit 3.0 Fluorometer (ThermoFisher Scientific) with the Qubit dsDNA HS Assay kit (ThermoFisher Scientific). https://doi.org/10.6084/m9.figshare.13487427.
(DOCX)

## Acknowledgments

We would like to thank Mr. Inoue, President of KITAZATO Corporation, for his dedication to the procurement of culture medium and other equipment. We would also like to thank Editage (www.editage.com) for English language editing.

## Author Contributions

**Conceptualization:** Akihiro Shitara, Kazumasa Takahashi, Hiromitsu Shirasawa, Yukihiro Terada.

**Data curation:** Akihiro Shitara, Kazumasa Takahashi.

**Formal analysis:** Akihiro Shitara, Kazumasa Takahashi.

**Funding acquisition:** Akihiro Shitara, Yukihiro Terada.

**Investigation:** Akihiro Shitara, Kazumasa Takahashi, Mayumi Goto.

**Methodology:** Akihiro Shitara, Kazumasa Takahashi.

**Project administration:** Akihiro Shitara, Kazumasa Takahashi, Yukihiro Terada.

**Resources:** Akihiro Shitara, Kazumasa Takahashi, Mayumi Goto.

**Software:** Akihiro Shitara, Kazumasa Takahashi.

**Supervision:** Akihiro Shitara, Kazumasa Takahashi.

**Validation:** Akihiro Shitara, Kazumasa Takahashi, Mayumi Goto.

**Visualization:** Akihiro Shitara, Mayumi Goto.

**Writing – original draft:** Akihiro Shitara, Kazumasa Takahashi, Yukihiro Terada.

**Writing – review & editing:** Akihiro Shitara, Kazumasa Takahashi, Harunori Takahashi, Takuya Iwasawa, Yohei Onodera, Kenichi Makino, Hiroshi Miura, Hiromitsu Shirasawa, Wataru Sato, Yukiyo Kumazawa, Yukihiro Terada.

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
