## [Decision Letter · Decision Letter 0]

1 Dec 2020

PONE-D-20-31386

Cell-free DNA in spent culture medium effectively reflects the chromosomal status of embryos following an extended culture compared to trophectoderm biopsy

PLOS ONE

Dear Dr. Shitara,

Thank you for submitting your manuscript to PLOS ONE. After careful consideration, we feel that it has merit but does not fully meet PLOS ONE’s publication criteria as it currently stands. Therefore, we invite you to submit a revised version of the manuscript that addresses the points raised during the review process.

We look forward to receiving your revised manuscript.

Kind regards,

Christine Wrenzycki

Academic Editor

PLOS ONE

2. In the Methods section, please provide the sequences of the specific primers used in the PCR analysis conducted in your study.

3. Please ensure you have thoroughly discussed any potential limitations of this study within the Discussion section, including the potential impact of confounding factors.

Reviewers' comments:

**C****omments to the Author**

1. Is the manuscript technically sound, and do the data support the conclusions?

Reviewer #1: Yes

Reviewer #2: Partly

2. Has the statistical analysis been performed appropriately and rigorously? 

Reviewer #1: Yes

Reviewer #2: Yes

3. Have the authors made all data underlying the findings in their manuscript fully available?

Reviewer #1: Yes

Reviewer #2: Yes

4. Is the manuscript presented in an intelligible fashion and written in standard English?

Reviewer #1: Yes

Reviewer #2: No

5. Review Comments to the Author

Reviewer #1: The authors have done a comparative study with donated human embryos to show if invasive vs. non-invasive methods perform best in detecting the aneuploidy status.

General comments

The study is well done and very interesting. Still, some points need clarification or are worth discussing.

1. The authors had the possibility to do their experiments with donated embryos. Even though the experiments are very well planned and performed, one has to remember that these were not first choice embryos. There was a reason why these embryos were not chosen for embryo transfer and rather cryopreserved (and the total number is still low). This detail can be seen in Table 2. Some embryos were not good, e.g. 3BB on day 6. This could explain probably also the chromosomal status.

2. They have removed the Zona using Tyrode solution. Could this probably also lead to cell death and consecutively to release of cfDNA from normal or affected cells?

3. The removal of the Zona is somehow artificial before TE biopsy. Normally it is not done this way. Usually a hole is drilled into the Zona with a laser. Removing the Zona also means that cells in the perivitelline space, which are excluded from the embryo, are probably lost due to washing steps (to remove the Tyrode). In reality (normal TE biopsy) these cells remain there in the embryo underneath the Zona and could possibly also release cfDNA which is then included in the analysis of the spent medium.

So this experiment does not completely reflect reality, although it is quite near.

4. The embryos were frozen on day 5 or 6 and all biopsied after warming on day 6. The authors state that spent medium was collected from recovery culture.

Does this mean that day 5 embryos were cultured for 24h and then biopsied and the 24h incubation spent medium was collected? And day 6 embryos were warmed and then put in recovery medium for how long? Maybe 2h? After that the TE biopsy was performed and then the spent medium was collected?

If the procedure was like this, was the short term incubation of day 6 embryos after thawing enough time to get cfDNA? Was there any difference among these two groups?

5. The authors state that in case of bad development or stop of growing the attached embryos were collected as a whole on day 8 or day 9, while from day 10 embryos the center of the attached embryo was biopsied and sampled. Additionally samples were taken from embryos that detached after being attached before. How can the authors be sure that equal numbers of cells or equal amounts of cell lineages (trophoblast or embryoblast) were in their samples? Unfortunately, this information cannot be found in table 2, but would be very important for the reader.

Was early detachment a sign of higher aneuploidy or mosaic status?

Lower quality attached embryos would have a higher amount of TE cells (or better descendent of TE cells) than good quality attached embryos. Would the number of trophoblast-originated cells have an influence on mosaic status?

Were all embryos attached at one time point in development? Data from Popovic et al. 2018 and from last ESHRE online meeting) indicated that embryos with multiple chromosomal defects do not attach at all. Why is it different in these embryos?

6. The amount of DNA recovered for NiPGT-A was with a lower range limit of 9.3ng/µL extremely low. How correct is the WGA in these cases? Could there be an amplification bias due to these low amounts?

7. The authors conclude their results (line 282) with stating that NiPGT-A had a trend to higher diagnostic accuracy. However, I would be more cautious as this was mainly true for day 10 outgrowths.

This trend was then formulated in the conclusion (line 380) as if it is completely supported by their results. As I said, I would be a bit more cautious with this very strong conclusion.

Minor details

- Line 67: 5-10 cells are normally retrieved (as described later by themselves)

- Please extend the legends of the tables so that the reader can understand the content without reading the text.

- Table 1, that do the authors mean with “CN”? Is there the “20” (number of samples) in PGT-A and Outgrowth in the first row missing?

- Table 2. What do the authors mean with “whole”? please add if outgrowth biopsy was a biopsy or whether the whole embryo was used for analysis.

- Line 217. “chromosomes” developed? Is there a word missing?

- Table 5. Please add in the legend of the table that the calculations were done with respect to outgrowth chromosomes

Reviewer #2: This study was undertaken to investigate the relative concordance of chromosome profiles obtained from non-invasive preimplantation genetic testing (ni-PGTA) and trophectoderm biopsy as compared with specimens obtained from embryos on days 8, 9 and 10 of culture. A total of 20 embryos were used, five of which were analyzed on day 8, five on day 9 and 10 on day 10. Based on the respective performance characteristics, the authors conclude that niPGT-A may be more accurate than trophectoderm biopsy for analysis of ploidy in the embryo outgrowths.

GENERAL COMMENTS

The authors should be commended for tackling this first study to investigate the relative efficacy of niPGT-A versus trophectoderm biopsy for predicting the chromosomal status of the early post-implantation human embryo. As such, this is a landmark study. However, the small number of embryos analyzed limits the ability to draw the conclusion that niPGT-A may be superior to trophectoderm biopsy for predicting chromosomal status of the embryo. In addition, as indicated below, there are several clarifications needed in the methods to ensure that a reader can reproduce the study accurately.

SPECIFIC COMMENTS

1) Nomenclature and acronyms: I strongly recommend that the authors adhere to the conventions used in the field as follows:

a. Trophectoderm biopsy should be referred to as such (the abbreviation “TE biopsy” may be used. Using “PGT-A” to refer to TE biopsy is confusing as “PGT-A” refers to the general test for aneuploidy and not the method by which this is achieved.

b. The acronym “dpf” for “days post fertilization” introduces unnecessary confusion. I recommend using the convention of “Day X”. i.e. day 8 embryos, day 9 embryos etc.

c. I recommend not abbreviating “spent culture medium” to SCM, but rather using the full term.

d. “extended culture” in the field of IVF, conventionally refers to culture of embryos up until day 5 or 6, and not beyond. I strongly recommend that you replace “extended culture” with “culture beyond implantation”

e. I suggest using “niPGT-A” rather than “NiPGT-A” as the former is the convention in the our field.

2) Materials and Methods

a. L. 124-126: Collection of the spent culture medium needs to clarified. As I understand it, 10 embryos were day 5 and 10 were day 6. Figure 1 indicates that the media were collected on day 6. Therefore, media collections occurred shortly after TE biopsy in half of the embryos (the day 6 group), whereas the collections occurred after ~24 hours of culture for the other 50% of embryos (day 5 embryos). Details regarding these specifics, whether the embryos were rinsed immediately post-biopsy and placed in fresh medium, the volume of the medium drop used etc. need to be included in the text. I also suggest that you delete the term “recovery culture”, which adds to the confusion of the methods used. Also, please note that there is a discrepancy in the description for when the spent media samples were collected – line 124 indicates this was done after the TE biopsy, whereas line 289 states this was the medium used from thawing to TE biopsy. Finally, the cfDNA concentrations for each of the day 5 and day 6 embryos separately should be reported in the Results (l. 188-190) and discussed in the Discussion on l. 352.

b. L. 140-149: It is stated on line 147 that “the center of the embryo was biopsied”. I assume that the sampling was, indeed, taken from the center of the embryo itself as opposed to the center of the outgrowth? If so, please emphasize this in the text and explain how contamination of TE cells was avoided during the collection. Further, reference to this sampling should be given a consistent name throughout the manuscript (it is currently referred to as “outgrowth (Table 1), “whole” (Table 2), “embryo” (title and running title), “outgrowth samples” (numerous locations in the text).

3) Results

a. Table 1: “Aneuploid” is spelt incorrectly

b. Table 2: The columns need to be adjusted so that the results all line up correctly with each embryo number

c. L. 213 and 217: “chromosomes” should read “embryos”

d. Table 3: The tiles need to be formatted correctly

e. Figure 1: “culture” is miss-spelt

f. L. 278: the FNR is missing for the niPGT-A group

4) Discussion

a. L. 296-304: The English here suggests a relatively high concordance rate between TE and ICM samples, yet on line 305, it is concluded that TE biopsies do not accurately reflect the ICM. Please reword to make the logic flow.

6. PLOS authors have the option to publish the peer review history of their article (what does this mean?). If published, this will include your full peer review and any attached files.

Reviewer #1: No

Reviewer #2: No

---

## [Author Response · Author response to Decision Letter 0]

30 Dec 2020

Response to all Reviewers

We greatly appreciate all the reviewers for their advice, which helped us improve the quality of the manuscript. We have revised the manuscript based on the reviewers’ comments. Our revisions in the manuscript are indicated with red text.

We have made the following major revisions: 

Responses to Academic Editor are provided below

1. We will revise as recommended.

2. We used a VeriSeq kit for PCR analysis in this study. This product uses random primers. Primer sequences are not provided on the product. 

3. We have explained the limitations of this study at the end of the discussion section. 

Responses to each reviewer are provided below.

Response to Reviewers #1

The authors have done a comparative study with donated human embryos to show if invasive vs. non-invasive methods perform best in detecting the aneuploidy status.

The study is well done and very interesting. Still, some points need clarification or are worth discussing..

Response:

Thank you for your comments. They were all very useful for improving the manuscript.

1) The authors had the possibility to do their experiments with donated embryos. Even though the experiments are very well planned and performed, one has to remember that these were not first choice embryos. There was a reason why these embryos were not chosen for embryo transfer and rather cryopreserved (and the total number is still low). This detail can be seen in Table 2. Some embryos were not good, e.g. 3BB on day 6. This could explain probably also the chromosomal status.

Response:

You are absolutely correct. The embryos used in this study were surplus embryos that were not used for transfers. There were some limitations restricting the embryos we could have used in the experiments: since human embryos are valuable, we could not use many of them.

We have included this point in the limitations at the end of the discussion section. (Line 379)

2) They have removed the Zona using Tyrode solution. Could this probably also lead to cell death and consecutively to release of cfDNA from normal or affected cells?

Response:

Zona removal using Tyrode is a method that has been used for a long time. At our center, embryos are carefully manipulated under a microscope by a skilled embryologist. We believe it is unlikely that Tyrode led to cell death.

3) The removal of the Zona is somehow artificial before TE biopsy. Normally it is not done this way. Usually a hole is drilled into the Zona with a laser. Removing the Zona also means that cells in the perivitelline space, which are excluded from the embryo, are probably lost due to washing steps (to remove the Tyrode). In reality (normal TE biopsy) these cells remain there in the embryo underneath the Zona and could possibly also release cfDNA which is then included in the analysis of the spent medium.

Response:

Certainly, there are some differences between our procedure and PGT-A as normally performed in clinical settings. In this study, we removed the zona pellucida to eliminate the effects of substances that could create measurement noise (e.g., spermatozoa and cumulus oophorus cells stuck to the Zona).

4) The embryos were frozen on day 5 or 6 and all biopsied after warming on day 6. The authors state that spent medium was collected from recovery culture.

Does this mean that day 5 embryos were cultured for 24h and then biopsied and the 24h incubation spent medium was collected? And day 6 embryos were warmed and then put in recovery medium for how long? Maybe 2h? After that the TE biopsy was performed and then the spent medium was collected?

If the procedure was like this, was the short term incubation of day 6 embryos after thawing enough time to get cfDNA? Was there any difference among these two groups?

Response:

We agree that the language here made our experimental procedure difficult to understand, as you pointed out. We have described it in greater detail. (Revise figure.) 

Incubation time until cfDNA collection differed between 5-day-old and 6-day-old embryos; however, there was no corresponding significant difference in DNA concentration. (Line 186-190.) 

5) The authors state that in case of bad development or stop of growing the attached embryos were collected as a whole on day 8 or day 9, while from day 10 embryos the center of the attached embryo was biopsied and sampled. Additionally samples were taken from embryos that detached after being attached before. How can the authors be sure that equal numbers of cells or equal amounts of cell lineages (trophoblast or embryoblast) were in their samples? Unfortunately, this information cannot be found in table 2, but would be very important for the reader.

Was early detachment a sign of higher aneuploidy or mosaic status?

Lower quality attached embryos would have a higher amount of TE cells (or better descendent of TE cells) than good quality attached embryos. Would the number of trophoblast-originated cells have an influence on mosaic status?

Were all embryos attached at one time point in development? Data from Popovic et al. 2018 and from last ESHRE online meeting) indicated that embryos with multiple chromosomal defects do not attach at all. Why is it different in these embryos?

Response:

Our method for collecting embryo samples on days 8 and 9 was different from that of day 10. However, it was not possible to accurately assess the corresponding differences in cell counts and lineages. Immunostaining would have been necessary to confirm cell lineages, but it was impossible to perform immunostaining considering the equipment available at our center since next-generation sequencing (NGS) analysis becomes impossible once immunostaining is performed. Since NGS analysis is not possible after immunostaining, the effects of trophoblast cell count on mosaic status would appear unknown.

We observed growth until day 8 in all embryos in our experiments. We considered an embryo “attached” if it anchored to the culture dish during the developmental stages and “detached” if it peeled off the dish after attachment. Embryos were recovered once detachment was confirmed.

6) The amount of DNA recovered for NiPGT-A was with a lower range limit of 9.3ng/µL extremely low. How correct is the WGA in these cases? Could there be an amplification bias due to these low amounts?

Response:

Concentrations of DNA obtained from samples using WGA were 9.3–32.3 ng/μl in the niPGT-A group. We have revised this sentence for clarity. (Lines 353-355) 

7) The authors conclude their results (line 282) with stating that NiPGT-A had a trend to higher diagnostic accuracy. However, I would be more cautious as this was mainly true for day 10 outgrowths.

This trend was then formulated in the conclusion (line 380) as if it is completely supported by their results. As I said, I would be a bit more cautious with this very strong conclusion.

Response:

We agree with you as this point is very important. We have softened the tone of the conclusion. 

8)Line 67: 5-10 cells are normally retrieved (as described later by themselves)

Response:

We have revised as suggested. (Lines 57-60) 

9) Please extend the legends of the tables so that the reader can understand the content without reading the text.

Line 10. the term cleavage rates is not a good term. It may refer to some sort of velocity. What might be a better term (except maybe for the first cleavage) is development rate or development stage. Fix throughout rest of document.

Response:

We have revised the table legends and figure as you suggested. 

10) Table 1, that do the authors mean with “CN”? Is there the “20” (number of samples) in PGT-A and Outgrowth in the first row missing?

Response:

CN = copy number. We forgot to define this abbreviation; we have fixed it throughout the manuscript. We have also revised Table 1. 

11) Table 2. What do the authors mean with “whole”? please add if outgrowth biopsy was a biopsy or whether the whole embryo was used for analysis.

Response:

This was a typographic error; we have fixed it. 

12) Line 217. “chromosomes” developed? Is there a word missing?

Response:

This was a diction error; we have fixed it. (Line 217) 

13)Table 5. Please add in the legend of the table that the calculations were done with respect to outgrowth chromosomes

Response:

We have revised the legend of Table 5, as suggested. (Line 261) 

Response to Reviewers #2

This study was undertaken to investigate the relative concordance of chromosome profiles obtained from non-invasive preimplantation genetic testing (ni-PGTA) and trophectoderm biopsy as compared with specimens obtained from embryos on days 8, 9 and 10 of culture. A total of 20 embryos were used, five of which were analyzed on day 8, five on day 9 and 10 on day 10. Based on the respective performance characteristics, the authors conclude that niPGT-A may be more accurate than trophectoderm biopsy for analysis of ploidy in the embryo outgrowths.

The authors should be commended for tackling this first study to investigate the relative efficacy of niPGT-A versus trophectoderm biopsy for predicting the chromosomal status of the early post-implantation human embryo. As such, this is a landmark study. However, the small number of embryos analyzed limits the ability to draw the conclusion that niPGT-A may be superior to trophectoderm biopsy for predicting chromosomal status of the embryo. In addition, as indicated below, there are several clarifications needed in the methods to ensure that a reader can reproduce the study accurately.

Response:

Thank you for your comments. We agreed with you, and your comments have helped us improve the manuscript.

1) Nomenclature and acronyms: I strongly recommend that the authors adhere to the conventions used in the field as follows:

a. Trophectoderm biopsy should be referred to as such (the abbreviation “TE biopsy” may be used. Using “PGT-A” to refer to TE biopsy is confusing as “PGT-A” refers to the general test for aneuploidy and not the method by which this is achieved.

Response:

We have revised this as recommended throughout the manuscript. 

b. The acronym “dpf” for “days post fertilization” introduces unnecessary confusion. I recommend using the convention of “Day X”. i.e. day 8 embryos, day 9 embryos etc.

Response:

We have revised this throughout the manuscript, as recommended. However, we used the “X-day-old” convention 

c. I recommend not abbreviating “spent culture medium” to SCM, but rather using the full term.

Response

We have used the full term throughout the manuscript, as recommended. 

d. “extended culture” in the field of IVF, conventionally refers to culture of embryos up until day 5 or 6, and not beyond. I strongly recommend that you replace “extended culture” with “culture beyond implantation”

Response:

We have revised this throughout the manuscript, as recommended. 

e. I suggest using “niPGT-A” rather than “NiPGT-A” as the former is the convention in the our field.

Response:

We have revised this throughout the manuscript, as recommended. 

2) Materials and Methods

a. L. 124-126: Collection of the spent culture medium needs to clarified. As I understand it, 10 embryos were day 5 and 10 were day 6. Figure 1 indicates that the media were collected on day 6. Therefore, media collections occurred shortly after TE biopsy in half of the embryos (the day 6 group), whereas the collections occurred after ~24 hours of culture for the other 50% of embryos (day 5 embryos). Details regarding these specifics, whether the embryos were rinsed immediately post-biopsy and placed in fresh medium, the volume of the medium drop used etc. need to be included in the text. I also suggest that you delete the term “recovery culture”, which adds to the confusion of the methods used. Also, please note that there is a discrepancy in the description for when the spent media samples were collected – line 124 indicates this was done after the TE biopsy, whereas line 289 states this was the medium used from thawing to TE biopsy. Finally, the cfDNA concentrations for each of the day 5 and day 6 embryos separately should be reported in the Results (l. 188-190) and discussed in the Discussion on l. 352.

Response:

Reviewer 1 had a similar suggestion. We have revised the experimental procedure, as well as the figure. We have also added the cfDNA concentrations. (Lines 184-188) 

b. L. 140-149: It is stated on line 147 that “the center of the embryo was biopsied”. I assume that the sampling was, indeed, taken from the center of the embryo itself as opposed to the center of the outgrowth? If so, please emphasize this in the text and explain how contamination of TE cells was avoided during the collection. Further, reference to this sampling should be given a consistent name throughout the manuscript (it is currently referred to as “outgrowth (Table 1), “whole” (Table 2), “embryo” (title and running title), “outgrowth samples” (numerous locations in the text).

Response:

Reviewer 1 had a similar suggestion. We have revised accordingly throughout the manuscript.

3) Results

a. Table 1: “Aneuploid” is spelt incorrectly

Response:

We have revised this as recommended. 

b. Table 2: The columns need to be adjusted so that the results all line up correctly with each embryo number

Response:

We have revised this as recommended. 

c. L. 213 and 217: “chromosomes” should read “embryos”

Response: We have revised this as recommended. (Lines 213, 217)

d. Table 3: The tiles need to be formatted correctly

Response:

We have revised this as recommended. 

e. Figure 1: “culture” is miss-spelt

We have revised this as recommended. 

f. L. 278: the FNR is missing for the niPGT-A group

We have provided the missing information. (Line 279)

4) Discussion

a. L. 296-304: The English here suggests a relatively high concordance rate between TE and ICM samples, yet on line 305, it is concluded that TE biopsies do not accurately reflect the ICM. Please reword to make the logic flow.

Response:

You are correct; we have revised the text. (Lines 295-304)

---

## [Decision Letter · Decision Letter 1]

20 Jan 2021

Cell-free DNA in spent culture medium effectively reflects the chromosomal status of embryos following culturing beyond implantation compared to trophectoderm biopsy

PONE-D-20-31386R1

Dear Dr. Shitara,

We’re pleased to inform you that your manuscript has been judged scientifically suitable for publication and will be formally accepted for publication once it meets all outstanding technical requirements.

Kind regards,

Christine Wrenzycki

Academic Editor

PLOS ONE

Additional Editor Comments (optional):

Reviewers' comments:

Reviewer's Responses to Questions

**Comments to the Author**

1. If the authors have adequately addressed your comments raised in a previous round of review and you feel that this manuscript is now acceptable for publication, you may indicate that here to bypass the “Comments to the Author” section, enter your conflict of interest statement in the “Confidential to Editor” section, and submit your "Accept" recommendation.

Reviewer #2: All comments have been addressed

2. Is the manuscript technically sound, and do the data support the conclusions?

Reviewer #2: (No Response)

3. Has the statistical analysis been performed appropriately and rigorously? 

Reviewer #2: (No Response)

4. Have the authors made all data underlying the findings in their manuscript fully available?

Reviewer #2: (No Response)

5. Is the manuscript presented in an intelligible fashion and written in standard English?

Reviewer #2: (No Response)

6. Review Comments to the Author

Reviewer #2: (No Response)

7. PLOS authors have the option to publish the peer review history of their article (what does this mean?). If published, this will include your full peer review and any attached files.

Reviewer #2: No

---

## [Editor Report · Acceptance letter]

28 Jan 2021

PONE-D-20-31386R1 

Cell-free DNA in spent culture medium effectively reflects the chromosomal status of embryos following culturing beyond implantation compared to trophectoderm biopsy 

Dear Dr. Shitara:

I'm pleased to inform you that your manuscript has been deemed suitable for publication in PLOS ONE. Congratulations! Your manuscript is now with our production department. 

Kind regards, 

on behalf of

Dr Christine Wrenzycki 

Academic Editor

PLOS ONE